# An insight into the causal relationship between sarcopenia-related traits and venous thromboembolism: A mendelian randomization study

Xinchao Du[1], Zhiwei Yao[2]*, Dongwei Wang[3], Xinwei Dong[1], Juncai Bai[3], Yingchun Gu[3], Yaohua Yu[4], Weifeng Zhang[5], Qingxia Qi[1], Shengyuan Gu[1]*

1 Department of Cardiology, Zhengzhou Central Hospital Affiliated to Zhengzhou University, Zhengzhou, Henan, China, 2 Department of Thyroid Surgery, The Affiliated Taian City Central Hospital of Qingdao University, Taian, Shandong, China, 3 Department of Cardiac Rehabilitation, Zhengzhou Central Hospital Affiliated to Zhengzhou University, Zhengzhou, Henan, China, 4 Department of Respiratory Medicine and Pulmonary Rehabilitation, Zhengzhou Central Hospital Affiliated to Zhengzhou University, Zhengzhou, Henan, China, 5 Department of Rheumatism and Immunology, Zhengzhou Central Hospital Affiliated to Zhengzhou University, Zhengzhou, Henan, China

* aspiring_yao@126.com (ZWY); gootionyuan@126.com (SYG)

**Data Availability Statement:** All data of this study are publicly available and can be obtained from the relevant GWAS articles and GWAS summary data sources. All the sources of the relevant data are

## Abstract

### Background

As a geriatric syndrome, sarcopenia has a high prevalence in the old population and represents an impaired state of health with adverse health outcomes. A strong clinical interest in its relationship with venous thromboembolism (VTE), which is a complex trait disease with a heterogeneous annual incidence rate in different countries, has emerged. The relationship between sarcopenia and venous thromboembolism has been reported in observational studies but the causality from sarcopenia to VTE remained unclarified. We aimed to assess the causal effect of sarcopenia on the risk of VTE with the two-sample Mendelian randomization (MR) method.

### Methods

Two sets of single-nucleotide polymorphisms (SNPs), derived from two published genome-wide association study (GWAS) meta-analyses and genetically indexing muscle weakness and lean muscle mass separately, were pooled into inverse variance weighted (IVW), weighted median and MR-Egger analyses.

### Results

No evidence was found for the causal effect of genetically predicted muscle weakness (IVW: $OR = 0.90$, 95% $CI = 0.76–1.06$, $p = 0.217$), whole body lean mass (IVW: $OR = 1.01$, 95% $CI = 0.87–1.17$, $p = 0.881$) and appendicular lean mass (IVW: $OR = 1.13$, 95% $CI = 0.82–1.57$, $p = 0.445$) on the risk of VTE. However, both genetically predicted whole-body lean mass and appendicular lean mass can causally influence diabetes mellitus (IVW of

showed within the manuscript and its Supporting Information files.

**Funding:** Z.Y. SDYWZGKCJHLH2023096 to Zhiwei Yao. The joint initiation project of scientific and technological innovation for Shandong Province medical staff. There is no URL for the above funder. No, this study was independent of any sponsors or funders.

**Competing interests:** The authors have declared that no competing interests exist.

whole-body lean mass: $OR = 0.87$, 95% $CI = 0.78–0.96$, $p = 0.008$; IVW of appendicular lean mass: $OR = 0.71$, 95% $CI = 0.54–0.94$, $p = 0.014$) and hypertension (IVW of whole-body lean mass: $OR = 0.92$, 95% $CI = 0.87–0.98$, $p = 0.007$; IVW of appendicular lean mass: $OR = 0.84$, 95% $CI = 0.73–0.96$, $p = 0.013$).

## Conclusions

Genetically predicted sarcopenia does not causally influence VTE directly, but it might still have an indirect effect on VTE incidence via diabetes mellitus and hypertension.

## 1. Introduction

Sarcopenia, named "flesh poverty" in Greek, is a progressive and generalized skeletal muscle disorder, having a pathologic condition of "muscle failure" with two typical clinical features—loss of muscle strength and muscle mass. Sarcopenia is associated with ageing—primary sarcopenia, or with a range of underlying causes, including nutritional disorders, inactivity, and any kind of diseases or iatrogenic conditions affecting skeletal muscle fitness—secondary sarcopenia [1, 2]. The overall prevalence of sarcopenia in adults 65 years and over is approximately 6–22% [3]. As a geriatric syndrome, sarcopenia represents an impaired state of health with adverse health outcomes (e.g. increased incidence of falls and fractures, disabilities, increased risk of death) [4]. Yet, whether sarcopenia causally increases the risk of venous thromboembolism (VTE) remains unclarified. To date, there are only two relevant studies [5, 6] powering concentration on this issue. To be specific, Gao et al. have found that sarcopenia was an independent predictor for VTE in bladder cancer patients undergoing radical cystectomy [6]. Meanwhile, Alwani et al. observed that sarcopenic patients with autologous head and neck free flap reconstruction had a significantly higher rate of venous thromboembolism [5]. However, due to the inherent defects of conventional designs, a retrospective observational study is unable to completely exclude the possibility of potential confounders, so whether sarcopenia plays a causal role in the onset of VTE is still unclear.

In sharp contrast to observational study designs, Mendelian randomization (MR) can overcome these potential biases such as confounders or reverse causation, and conclude a reliable causal association of risk factors with disease outcome [7, 8]. In this study, we implemented an MR design, using large-scale genome-wide association study (GWAS) data of sarcopenia-related traits—muscle weakness and lean muscle mass—and VTE, to assess the causal effect of sarcopenia on the risk of VTE.

## 2. Methods

No additional ethics approval was needed due to the re-analysis of previously collected and published data.

### 2.1. Study design

The MR framework is shown in Fig 1. Overall, this two-sample MR analysis is not to violate three assumptions: (i) genetic instruments—index SNPs—predict the sarcopenia-related traits at a genome-wide significance level ($p < 5 \times 10^{-8}$) and are independent of VTE ($p > 5 \times 10^{-8}$); (ii) index SNPs are independent of potential confounders (e.g. VTE-related risk factors); (iii)

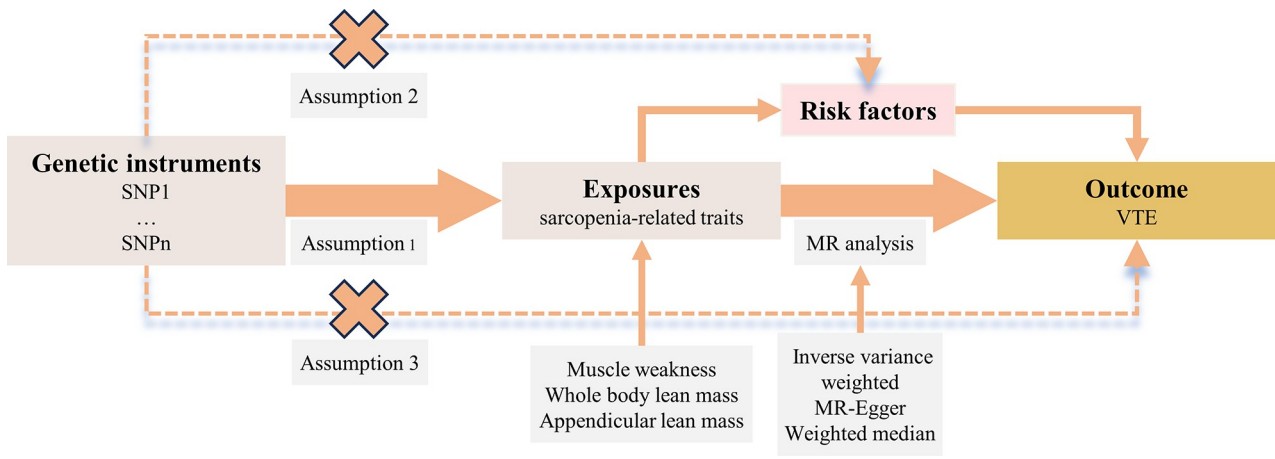

**Fig 1. The Mendelian randomization framework.**

index SNPs affects the VTE only via sarcopenia-related traits or VTE-related risk factors [9, 10].

## 2.2. Genetic variants associated with sarcopenia-related traits

Muscle weakness and lean muscle mass, which are the two agreed-upon key components running through various sets of diagnostic criteria for sarcopenia [2], were used as sarcopenia-related traits in this MR analysis.

**2.2.1. Muscle weakness-index SNPs.** The muscle weakness index SNPs were derived from a recent genome-wide meta-analysis [11]. Adjusted for age, sex, and technical covariates, 15 SNPs were reported significantly related to muscle weakness ($p < 5 \times 10^{-8}$). According to the phenotypes of the 15 SNPs summarized by Jones et al. [11], 7 SNPs were excluded due to their association with obesity and autoimmune disease which are VTE-related risk factors (S1 Table in S1 File). Another potential pleiotropy assessment of the rest 8 SNPs on VTE-related risk factors [12] was performed by searching human genotype-phenotype associations at a genome-wide level ($p < 5 \times 10^{-8}$) with the website tool PhenoScanner V2 (http://www.phenoscanner.medschl.cam.ac.uk/) [13, 14]. Details of the rest 8 SNPs, including the results of potential pleiotropy assessment, are shown in S2 Table in S1 File. To avoid violation of the first assumption of two-sample MR [10], the linkage disequilibrium (LD, $r^2 < 0.01$ and within 10000 kb) was evaluated by the package TwoSampleMR (version 0.5.7) in R (version 4.3.1) to remove unqualified SNPs before retrieving SNPs from outcome GWAS. The $F$-statistics of each of the eligible 8 SNPs was $> 10$ (S2 Table in S1 File), indicating that the SNPs had excellent potential for predicting muscle weakness [15].

**2.2.2. Lean muscle mass-index SNPs.** The set of SNPs indexing lean muscle mass was obtained from a GWAS meta-analysis [16]. After discovery and replication, and adjusting for sex, age, height and fat mass, the combined analysis only for European ancestry has confirmed 5 SNPs indexing predicted whole body lean mass in 85,519 participants and 3 SNPs indexing predicted appendicular lean mass in 70,690 participants. A potential pleiotropy assessment was also performed as described in the 'Muscle weakness-index SNPs' part under the 'Methods' section. Certainly, as described above, the linkage disequilibrium was also assessed. The eligible SNPs for being retrieved from outcome GWAS were shown in S3, S4 Tables in S1 File. The $F$-statistics of each SNP was $> 10$, which manifested that the SNPs had a strong potential for predicting lean muscle mass.

## 2.3. GWAS summary data for VTE

The FinnGen study is a large-scale genomics initiative that has analyzed over 500,000 Finnish biobank samples and correlated genetic variation with health data to understand disease mechanisms and predispositions [17]. The GWAS summary data for VTE were obtained from the FinnGen study. In total, there were 21,021 VTE cases and 391,160 control cases in FinnGen Data Freeze 10 that is released in December 2023.

## 2.4. Mendelian randomization analyses

After harmonizing the effect alleles between sarcopenia-related traits and VTE and removing the VTE-related SNPs (at a significance level of $p < 5 \times 10^{-8}$) (results of harmonization are shown in S5-S7 Tables in S1 File), three MR analytical methods, including inverse variance weighted (IVW), weighted median and MR-Egger, were used to assess the causal effects of genetically predicted sarcopenia-related traits on VTE. We took the standard random-effect IVW approach as the main analysis, and then the weighted median and the MR-Egger as the essential sensitivity analyses, which could make the estimates more robust from different scenarios because of the different assumptions underlying the three methods. Specifically, IVW, in which heterogeneity is acceptable, is more efficient than weighted median and MR-Egger to yield a narrower confidence interval (CI) due to the underlying assumption that all the variable instruments make an effect on the outcome only via the exposure, or several variable instruments have horizontal pleiotropy of other paths which is however balanced overall. Different from IVW, the principle of weighted median stands up upon that at least half of the variables are valid via the exposure, while MR-Egger could still work even if none of the variables is valid via the exposure [18, 19]. Due to the multiple MR approaches, if inconsistent results were driven by different methods, the exposure-related *p*-value threshold was tightened till a consistent result.

## 2.5. Sensitivity analyses

Sensitivity analysis, which assesses the robustness of the results and potential horizontal pleiotropy that could violate the two-sample MR framework assumption, includes Cochran's *Q* statistic, funnel pot, leave-one-out (LOO) analyses and MR-Egger intercept.

## 2.6. Risk factors

A deep and well-acceptable understanding about VTE is that the interaction of patient-related risk factors—a "bomb", usually permanent—and setting-related risk factors—the "trigger", usually temporary—leads to the resultant VTE [12]. According to this insight into the predisposing factors for VTE, after screening the items of risk factors [12], several patient-related ones, including heart failure (HF), atrial fibrillation (AF), myocardial infarction (MI), diabetes mellitus (DM), hypertension (HTN) and body mass index (BMI), were regarded as the potential mediators that might genetically link sarcopenia to VTE.

We obtain the GWAS summary data of AF, MI, DM and BMI respectively from Atrial Fibrillation Genetics (AFGen) consortium [20], Coronary Artery Disease Genome wide Replication and Meta-analysis (CARDIoGRAM) plus The Coronary Artery Disease (C4D) Genetics (CARDIoGRAMplusC4D) consortium [21], Diabetes Genetics Replication and Meta-analysis (DIAGRAM) consortium [22], and Genetic Investigation of Anthropometric Traits (GIANT) consortium [23]. GWAS summary data for HF and HTN were derived from the FinnGen study [17]. Details of risk factor data sources are shown in S8 Table in S1 File.

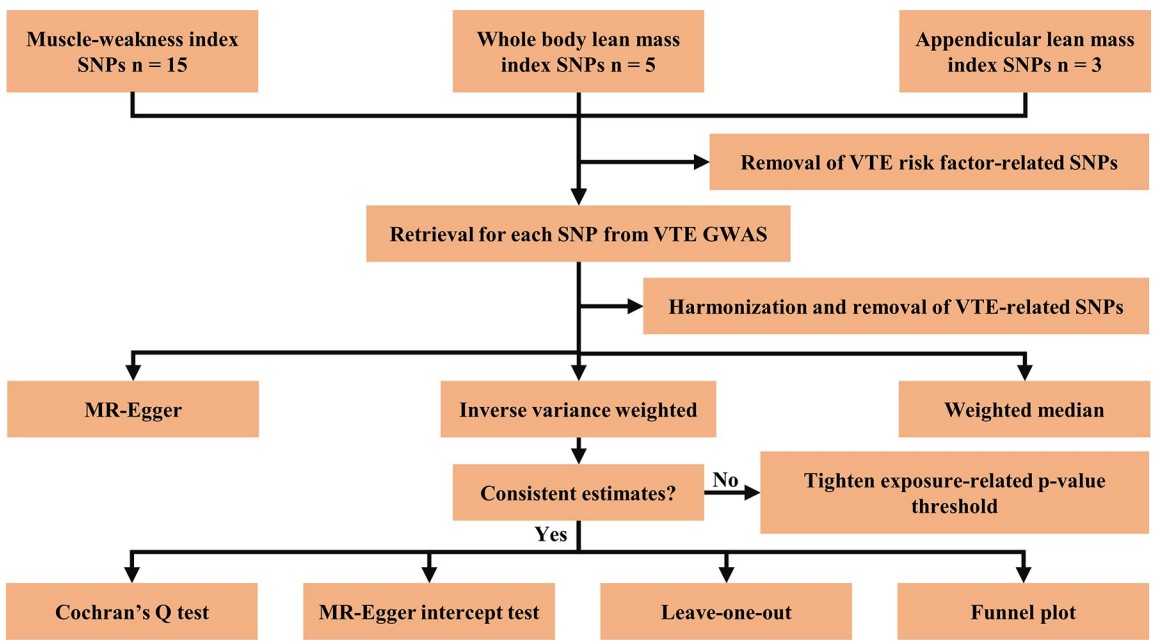

**Fig 2. The analysis flow chart of the Mendelian randomization study.**

Two-sample MR analyses were performed to assess the causal effect of sarcopenia on HF, AF, MI, DM, HTN and BMI, with the IVW method. As the secondary MR analyses, a two-tailed $p < 0.05$ was considered statistically significant without the Bonferroni correction.

## 2.7. Statistical analysis

A Bonferroni-corrected $p$ threshold of $< 0.0166$ ($\alpha = 0.05/3$ outcomes) was set to account for multiple testing in our primary analyses and in the meantime, $p < 0.05$ was considered to be nominally significant. *OR* with corresponding 95% *CI* were shown as the MR results. All analyses were performed with the package TwoSampleMR (version 0.5.7) in R (version 4.3.1). The analysis flow chart of the Mendelian randomization study is shown in Fig 2.

# 3. Results

## 3.1. Causal effect from muscle weakness to VTE

No statistically significant evidence was found for the causal effect of genetically predicted muscle weakness on the risk of VTE. IVW (*OR* = 0.90, 95% *CI* = 0.76–1.06, $p = 0.217$), MR-Egger (*OR* = 1.00, 95% *CI* = 0.62–1.60, $p = 0.997$) and weighted median (*OR* = 0.86, 95% *CI* = 0.68–1.09, $p = 0.203$) showed a consistent result with each other (Table 1 and S1 Fig in

**Table 1. Causal effects for sarcopenia-related traits on VTE.**

| Exposure | Outcome | IVW | | | MR-Egger | | | WM | | |
|---|---|---|---|---|---|---|---|---|---|---|
| | | *OR* | 95% *CI* | *p* | *OR* | 95% *CI* | *p* | *OR* | 95% *CI* | *p* |
| muscle weakness | VTE | 0.90 | 0.76–1.06 | 0.217 | 1.00 | 0.62–1.60 | 0.997 | 0.86 | 0.68–1.09 | 0.203 |
| whole body lean mass | VTE | 1.01 | 0.87–1.17 | 0.881 | 0.79 | 0.62–1.02 | 0.321 | 0.99 | 0.86–1.14 | 0.859 |
| appendicular lean mass | VTE | 1.13 | 0.82–1.57 | 0.445 | NA | NA | NA | NA | NA | NA |

IVW: inverse variance weighted; WM: weighted median; NA: not applicable to MR-Egger and WM analyses.

**Table 2. Sensitivity analysis of the causal association between sarcopenia-related traits and the risk of VTE.**

| Exposure | Outcome | Cochran Q test | | MR-Egger | |
|---|---|---|---|---|---|
| | | *Q* value | *p* | *Intercept* | *p* |
| muscle weakness | VTE | 5.24 | 0.513 | -0.006 | 0.660 |
| whole body lean mass | VTE | 4.36 | 0.113 | 0.044 | 0.287 |
| appendicular lean mass | VTE | 1.88 | 0.170 | NA | NA |

NA: not applicable to the MR-Egger intercept test.

S2 File). No heterogeneity was observed with a Cochran *Q*-test derived from the IVW method (*p* = 0.513). No directional pleiotropy was detected with an MR-Egger-regression intercept test (*intercept* = -0.006, *se* = 0.013, *p* = 0.660) (Table 2). LOO sensitivity analysis also showed that MR results were not driven by any single SNP, indicating the robustness of the analyses (S2 Fig in S2 File). The funnel plot is shown in S3 Fig in S2 File.

### 3.2. Causal effect from lean muscle mass to VTE

**3.2.1. Causal effect from whole body lean mass to VTE.** MR analyses presented no evidence for the causal effect of genetically predicted whole body lean mass on VTE, from the similar results from IVW (*OR* = 1.01, 95% *CI* = 0.87–1.17, *p* = 0.881), MR-Egger (*OR* = 0.79, 95% *CI* = 0.62–1.02, *p* = 0.321) and weighted median (*OR* = 0.99, 95% *CI* = 0.86–1.14, *p* = 0.859) (Table 1 and S4 Fig in S2 File). A Cochran *Q*-test with a *p* value of 0.113 showed no heterogeneity existing in these MR analyses. An MR-Egger regression intercept-test (*intercept* = 0.044, *se* = 0.021, *p* = 0.287) presented no directional pleiotropy violating the two-sample MR framework (Table 2). LOO analysis showed robustness for these analyses (S5 Fig in S2 File).

**3.2.2. Causal effect from appendicular lean mass to VTE.** SNP rs4842924 and rs2287926, significantly associated with appendicular lean mass, were identified independent of VTE and VTE risk factors, and then only the IVW method was performed. The result of IVW (*OR* = 1.13, 95% *CI* = 0.82–1.57, *p* = 0.445) indicated that genetically predicted appendicular lean mass had no causality on VTE (Table 1). No obvious heterogeneity was observed in the Cochran-*Q* test (*p* = 0.170) (Table 2).

Due to the negative MR results being consistent across multiple MR approaches (as shown in the two parts under the 'Results' section—'Causal effect from muscle weakness to VTE' and 'Causal effect from lean muscle mass to VTE'), the exposure-related *p*-value threshold was not tightened as described in the 'Methods' section and analysis flow chart.

### 3.3. Causal effect from sarcopenia-related traits on VTE risk factors

The MR estimates showed that genetically predicted sarcopenia-related traits had no causal effect on the risk of VTE, but, nevertheless, we still assessed the causal effect of sarcopenia-related traits on VTE risk factors, to provide a broader insight into the relationship between sarcopenia and VTE. There were causal effects observed from lean muscle mass (whole-body lean mass and appendicular lean mass) to both DM and HTN (Table 3).

## 4. Discussion

This two-sample MR study was designed to reveal whether sarcopenia causally influenced VTE incidence and the MR estimates gave a negative conclusion that genetically predicted sarcopenia-related traits had no causal effect on the risk of VTE. Nonetheless, MR analyses for

**Table 3. Mendelian randomization estimates of IVW for the associations from sarcopenia-related traits to VTE risk factors.**

| Exposure | Outcome | Causal effect (95% *CI*) | *p* value |
|---|---|---|---|
| Muscle weakness | Heart failure | 0.91 (0.78–1.06) | 0.206 |
| | Atrial fibrillation | 1.02 (0.78–1.33) | 0.906 |
| | Myocardial infarction | 1.02 (0.78–1.34) | 0.889 |
| | Diabetes mellitus | 0.86 (0.60–1.24) | 0.428 |
| | Hypertension | 1.01 (0.92–1.12) | 0.790 |
| | Body mass index | 1.01 (0.86–1.18) | 0.932 |
| Whole body lean mass | Heart failure | 1.05 (0.95–1.16) | 0.351 |
| | Atrial fibrillation | 0.94 (0.84–1.06) | 0.311 |
| | Myocardial infarction | 0.91 (0.72–1.15) | 0.441 |
| | Diabetes mellitus | 0.87 (0.78–0.96) | 0.008* |
| **Exposure** | **Outcome** | **Causal effect (95% *CI*)** | ***p* value** |
| | Hypertension | 0.92 (0.87–0.98) | 0.007* |
| | Body mass index | 1.02 (0.98–1.05) | 0.307 |
| Appendicular lean mass | Heart failure | 1.10 (0.78–1.55) | 0.569 |
| | Atrial fibrillation | 0.84(0.64–1.10) | 0.213 |
| | Myocardial infarction | 0.84 (0.53–1.34) | 0.465 |
| | Diabetes mellitus | 0.71 (0.54–0.94) | 0.014* |
| | Hypertension | 0.84 (0.73–0.96) | 0.013* |
| | Body mass index | 1.05 (0.97–1.13) | 0.232 |

*: $p < 0.05$.

the association of sarcopenia with VTE-related risk factors suggested that genetically predicted sarcopenia-related traits had a causal effect on DM and HTN that could increase the incidence of VTE.

## 4.1. VTE and related predisposing factors

VTE, clinically presenting as deep vein thrombosis (DVT) and/or pulmonary embolism (PE), is a common and lethal pathophysiological condition [24] with a heterogeneous annual incidence rate in different countries ranging from 8–184 per 100 000 people [25], and heavily increases a global healthcare burden [26]. A valid method established by Virchow's triad to represent the pathophysiology of VTE includes blood flow stasis, hypercoagulability, endothelial dysfunction and injury [27], most VTE events are unprovoked though [28]. Nevertheless, there is still a range of well-recognized predisposing factors for VTE [12, 29, 30]. According to their diverse susceptibility to VTE, these identified risk factors are classified into three categories—strong (*OR* > 10), moderate (*OR* 2–9) and weak (*OR* < 2) [29, 30].

Undoubtedly, the pathogenesis of VTE is complex. Nonetheless, the higher the risk factors' *OR*, the more definite the pathogenesis from risk factors to VTE. A comprehensive insight into the relationship between VTE and weak/common risk factors should be based on a deep understanding of VTE. VTE may be regarded as part of the cardiovascular disease continuum, which is associated with arterial disease, especially atherosclerosis [31–33], by sharing weak/common risk factors (e.g. obesity, HTN, DM) [34]. However, this relationship between VTE and arterial disease may be indirect and mediated, at least in part, by the complications of coronary artery disease [12]. As an example, MI and HF increase the risk of PE [35]. In a word, weak/common risk factors (e.g. obesity, HTN, DM) increase VTE incidence probably by causing several complications of coronary artery disease—strong risk factors (e.g. MI, HF). This

view about VTE-related risk factors forms the foundation for our study of detecting the appropriate mediators, including MI, HF, AF, HTN, DM and BMI, which potentially link sarcopenia to VTE.

## 4.2. Superficial effect of sarcopenia on VTE

As is well known, sarcopenia and VTE have several common risk factors, including increasing age, immobility, bed rest, heart failure, cancer, etc. [1, 12], which easily implies in theory that in a specific case, for example for an old patient with a sedentary lifestyle, there might be a higher risk for both sarcopenia and VTE due to a common pathogenic mechanism from the sedentary lifestyle and ageing. Thus, it deserves to have the foresight that sarcopenia and VTE would accompany each other in complex clinical situations. According to this theoretical deduction and given the fact that this MR study gave a 'no' answer to the causal effect of sarcopenia on VTE, the significantly positive association of sarcopenia and VTE in the two retrospectively designed studies [5, 6] might be driven by potential confounders, such as some potential common pathogenic mechanisms. Specifically, in both studies [5, 6], the patients in the sarcopenia group were older than the non-sarcopenia group at a significant *p*-level, which indicated that it might be ageing but not sarcopenia that led to a higher VTE incidence in the sarcopenia group. Of course, we are convinced that there must be other undiscovered confounders, such as the potential difference in the degree of severity of cancer between sarcopenia and non-sarcopenia groups, attributing to these "false" positive results on the relationship between sarcopenia and VTE.

## 4.3. causal effect of sarcopenia on VTE-related risk factors

MR estimates of this study for assessing the causal association of sarcopenia with VTE-related risk factors gave sound evidence that genetically predicted sarcopenia increased the risk of DM and HTN.

A lot of cross-sectional studies have observed that low skeletal muscle mass had an association with DM [36–39]. About 75% of glucose disposal is taken by skeletal muscle, a primary organ responsible for insulin-stimulated glucose uptake [40], which means that sarcopenia patients have a dysfunctional glucose uptake from the circulation. What is more, skeletal muscle insulin resistance plays a key role for old sarcopenic people in developing DM [41]. More specifically speaking, accompanied by a loss of lean muscle mass, sarcopenic muscle has a decline in muscle quality, including reduced oxidative capacity and increased production of reactive oxygen species which leads to pro-inflammatory processes and oxidative mitochondrial DNA damage. Mitochondrial dysfunction and inflammation cause skeletal muscle-related insulin resistance [42, 43]. In line with the previous observational studies [36–39], we found that both genetically predicted whole-body lean mass and appendicular lean mass had a causal relation with DM.

Given the high prevalence of both sarcopenia and HTN in the old population [3, 44], several observational studies have explored the relationship between sarcopenia and HTN [45–47]. A meta-analysis by Bai et al. has found that sarcopenia was associated with hypertension, but no correlation was found between handgrip strength and hypertension [48]. Consistent with these previous conventional studies, we gave a genetic conclusion that both genetically predicted whole body lean mass and appendicular lean mass increased the risk of HTN and muscle weakness—low handgrip strength—had no causal correlation with HTN. The common biological mechanisms in sarcopenia, including oxidative stress, chronic low-grade systemic inflammation, mitochondrial dysfunction, etc., may account for the causal relation of sarcopenia with HTN [2].

Although there was no evidence supporting that genetically predicted sarcopenia causally influenced VTE directly, based on the premise, as already discussed or proved, that DM and HTN are the common/weak risk factors increasing VTE incidence by causing several complications of coronary artery disease—strong risk factors (e.g. MI, HF) and sarcopenia could genetically cause DM or HTN, it is sensible to hold a foresight that genetically predicted sarcopenia might have an indirect effect on VTE incidence via DM and HTN. By pooling the relevant literature into a comprehensive consideration [1, 2, 25, 43], it is easy to find a clue that chronic systemic inflammation plays a common and key role between sarcopenia, DM, HTN and VTE, which would provide a solid foundation for our inference and offer an insight into the biological pathways linking sarcopenia-related traits to DM and/or HTN and ultimately to VTE. Yet, these potential biological mechanisms underlying the observed indirect effects of sarcopenia on VTE through DM and HTN need more and further basic research to strengthen.

### 4.4. Strength

This MR study has several strengths. First, the application of MR design enables us to simulate RCT, which is widely accepted to explore causality, at a much larger sample-size level and in a more advantageous way that is free from ethical restriction, impracticality of implementation and confounding factors. Second, compared with observational studies which can only make a superficial association, MR can give a definite causal conclusion against reverse causal effects and confounders. In fact, it seems never possible to use conventional studies, either RCTs or observational studies, to detect a clear causal relationship of sarcopenia with VTE due to the complex and heterogeneous clinical background of recruited patients. Third, our findings may provide a new perspective on sarcopenia and VTE, and influence public health care policies (e.g. early prevention, timely intervention). The findings of our study implied that sarcopenia might have an indirect effect on VTE incidence via DM and HTN, and addressing early screening and timely intervening DM and HTN could perhaps be beneficial for patients with genetically predicted sarcopenia to reduce the risk of VTE.

### 4.5. Limitation

Our study also has several limitations. First, the GWAS data were all obtained from the European population, which restricts the range of application of our findings to other ethnicity. It is necessary to investigate whether our findings were robust in other races. Second, we have not detected the causal association of sarcopenia with lower limb fracture and hip or knee replacement which are also the major risk factor for VTE. Third, more attention should be paid to the difference between DVP and PE, two subtypes of VTE. Sarcopenia might have causality on a certain subtype of VTE. A broader study containing subgroups of DVP and PE can be considered in the future. Third, the *p*-value threshold of the secondary MR analyses of from sarcopenia to the VTE risk factors was not adjusted for multiple testing using the Bonferroni-corrected method as the primary analyses, which would increase the risk for the type I error, also the so-called "false positive", in statistics. After the Bonferroni correction (0.05/18), none of the MR estimates could reach a significant level (Table 3). Thus, our positive MR results about sarcopenia and the VTE risk factors should be treated cautiously, which means that if you accept our conclusions, you would risk a higher type I error in statistics than usual.

### 5. Conclusion

This is the first MR study to explore the causality from sarcopenia to VTE. We found that genetically predicted sarcopenia does not causally influence VTE directly, but it might still

have an indirect effect on VTE incidence via DM and HTN. Addressing early screening and timely intervening DM and HTN might be beneficial for patients with genetically predicted sarcopenia to reduce the risk of VTE.

## Supporting information

**S1 File. The Supporting information Table file containing S1-S8 Tables.** (XLSX)

**S2 File. The Supporting information Figure file containing S1-S5 Figs.** (DOCX)

## Acknowledgments

We want to acknowledge the participants and investigators of the FinnGen study. We also would like to acknowledge the participants and investigators of the AFGen, CARDIoGRAM-plusC4D, DIAGRAM and GIANT consortium.

## Author Contributions

**Conceptualization:** Xinchao Du, Shengyuan Gu.

**Formal analysis:** Xinchao Du, Zhiwei Yao.

**Funding acquisition:** Zhiwei Yao.

**Investigation:** Xinchao Du.

**Methodology:** Xinchao Du, Zhiwei Yao, Dongwei Wang.

**Project administration:** Xinchao Du.

**Resources:** Xinchao Du, Juncai Bai, Yaohua Yu, Weifeng Zhang.

**Software:** Xinchao Du.

**Supervision:** Shengyuan Gu.

**Validation:** Xinchao Du.

**Visualization:** Xinchao Du, Zhiwei Yao.

**Writing – original draft:** Xinchao Du.

**Writing – review & editing:** Zhiwei Yao, Dongwei Wang, Xinwei Dong, Juncai Bai, Yingchun Gu, Yaohua Yu, Weifeng Zhang, Qingxia Qi, Shengyuan Gu.

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
