## [Decision Letter · Decision Letter 0]

1 Apr 2024

PONE-D-24-01767An Insight into the Causal Relationship between Sarcopenia-Related Traits and Venous Thromboembolism: A Mendelian Randomization StudyPLOS ONE

Dear Dr. Gu,

Thank you for submitting your manuscript to PLOS ONE. After careful consideration, we feel that it has merit but does not fully meet PLOS ONE’s publication criteria as it currently stands. Therefore, we invite you to submit a revised version of the manuscript that addresses the points raised during the review process.

We look forward to receiving your revised manuscript.

Kind regards,

Bo Hu, PhD

Academic Editor

PLOS ONE

3. Thank you for stating the following financial disclosure: "Z.Y.

SDYWZGKCJHLH2023096 to Zhiwei Yao.

The joint initiation project of scientific and technological innovation for Shandong Province medical staff.

There is no URL for the above funder. 

No, this study was independent of any sponsors or funders." 

Additional Editor Comments:

This study was carefully conducted and the paper was well written. My main comment is regarding further clarifications needed for certain important methods, e.g., whether the analysis on the VTE risk factors was corrected for multiplicity.

Reviewers' comments:

Reviewer's Responses to Questions

**Comments to the Author**

1. Is the manuscript technically sound, and do the data support the conclusions?

Reviewer #1: Yes

Reviewer #2: Yes

2. Has the statistical analysis been performed appropriately and rigorously? 

Reviewer #1: Yes

Reviewer #2: Yes

3. Have the authors made all data underlying the findings in their manuscript fully available?

Reviewer #1: Yes

Reviewer #2: Yes

4. Is the manuscript presented in an intelligible fashion and written in standard English?

Reviewer #1: Yes

Reviewer #2: Yes

5. Review Comments to the Author

Reviewer #1: Title: An Insight into the Causal Relationship between Sarcopenia-Related Traits and Venous Thromboembolism: A Mendelian Randomization Study

The paper presents a meticulously executed Mendelian randomization study delving into the causal relationship between sarcopenia-related traits and VTE. The study design is robust, and the analyses are conducted with a high level of rigor. These findings significantly contribute to our comprehension of the intricate interplay between sarcopenia, VTE, and associated risk factors. However, there are opportunities for minor enhancements such as providing effect sizes for indirect effects, discussing study limitations more comprehensively, and offering more specific recommendations for future research and clinical practice. Nevertheless, this study constitutes a substantial addition to the literature on sarcopenia and VTE.

Abstract:

The abstract succinctly outlines the study's objective, methodology, key findings, and conclusions, effectively emphasizing the knowledge gap concerning the causality between sarcopenia and VTE. To enhance clarity and contextualize the study's significance, incorporating a brief discussion on the prevalence and clinical significance of sarcopenia and VTE would be beneficial. Additionally, a minor grammatical error is noted in the last sentence of the conclusion, where "did not" should be replaced with "does not" to ensure consistency in verb tense.

Methods:

The methods section is meticulously detailed and well-organized, providing clear explanations of the study design, genetic variant selection, data sources, and statistical analyses. The rationale behind the selection of sarcopenia-related traits, genetic variants, and outcome measures is well-justified. Commendably, sensitivity analyses are employed to assess the robustness of the results.

The sentence "After harmonization of the effect alleles between sarcopenia-related traits and VTE and removal of the VTE-related SNPs (at a level of p < 5 × 10−8)" could be improved to "After harmonizing the effect alleles between sarcopenia-related traits and VTE and removing the VTE-related SNPs (at a significance level of p < 5 × 10−8)".

While the paper provides a clear explanation of the MR methods used, including IVW, weighted median, and MR-Egger analyses, further elaboration on why these specific methods were chosen and how they complement each other in addressing potential biases could be beneficial.

Discussion and Conclusion:

Interpretation of Findings: The discussion section meticulously interprets the study's findings, particularly emphasizing the absence of direct causal evidence between sarcopenia-related traits and VTE. However, delving deeper into potential mechanisms underlying the observed indirect effects of sarcopenia on VTE through diabetes mellitus (DM) and hypertension (HTN) would strengthen this section. Offering insights into the biological pathways or physiological processes linking sarcopenia-related traits to these risk factors and ultimately to VTE would enhance the discussion.

Implications and Future Directions: The conclusion effectively summarizes the study's key findings and highlights the potential indirect effects of sarcopenia on VTE through DM and HTN. Expanding on the implications of these findings for clinical practice and future research directions would provide a more comprehensive closure to the paper. Discussing how these results could inform preventive strategies or interventions targeting sarcopenia-related traits to reduce the risk of VTE could add practical value to the study.

Reviewer #2: This manuscript, “An Insight into the Causal Relationship between Sarcopenia-Related Traits and Venous Thromboembolism: A Mendelian Randomization Study”, by Du et al introduced a Mendelian Randomization (MR) study to assess the causal effect of sarcopenia on the risk of VTE. While the results show that the SNPs genetically predicted sarcopenia did not causally influence VTE directly, this is a well-designed and interesting study. The paper was well written in general. I have several comments for the authors.

1. The secondary analyses of the VTE risk factors were not adjusted for multiple testing (Table 3). Please clarify that in the paper.

2. The method of tightening the p-values from different methods is vague. Please provide more explanations and illustrate the use in the results.

3. Minor comments:

a. Figure 1. Please add the number of SNP used at each step.

b. Section 2.4. The weight median and MR-Egger analyses are essentially sensitivity analyses but not complements.

6. PLOS authors have the option to publish the peer review history of their article (what does this mean?). If published, this will include your full peer review and any attached files.

Reviewer #1: No

Reviewer #2: No

---

## [Author Response · Author response to Decision Letter 0]

4 Apr 2024

Dear Editor Bo Hu,

 We would like to express our great appreciation to you and the volunteer reviewers.

 Thank you for the review comments, each point of which is helpful, constructive and meticulous. Although time is so precious, you spend your time processing and reviewing this article for a rigorous publication, it is self-giving of you! Humanity has made inexhaustible progress in the knowledge of nature, just because of anyone like you who works just for science, not for interest. As the co-corresponding author, I feel gratified because our scrupulousness won us plaudits from you, regardless of whether the review comments were positive. I have a “stubborn” belief that process counts more than results and attitude toward honesty and meticulousness counts more than ability. I can read your valuable quality of meticulousness and erudition from the meticulous review comments. And we are very glad to perfect our manuscript with your valuable peer-review suggestion. 

Thank you, you deserve it, all of you!

Yours sincerely,

Shengyuan Gu on behalf of the authors. (Zhiwei Yao and Shengyuan Gu as the co-corresponding authors)

---

## [Decision Letter · Decision Letter 1]

22 Apr 2024

An Insight into the Causal Relationship between Sarcopenia-Related Traits and Venous Thromboembolism: A Mendelian Randomization Study

PONE-D-24-01767R1

Dear Dr. Gu,

We’re pleased to inform you that your manuscript has been judged scientifically suitable for publication and will be formally accepted for publication once it meets all outstanding technical requirements.

Kind regards,

Bo Hu, PhD

Academic Editor

PLOS ONE

Additional Editor Comments (optional):

The authors have successfully addressed the previous comments

Reviewers' comments:

Reviewer's Responses to Questions

**Comments to the Author**

1. If the authors have adequately addressed your comments raised in a previous round of review and you feel that this manuscript is now acceptable for publication, you may indicate that here to bypass the “Comments to the Author” section, enter your conflict of interest statement in the “Confidential to Editor” section, and submit your "Accept" recommendation.

Reviewer #1: All comments have been addressed

Reviewer #2: (No Response)

2. Is the manuscript technically sound, and do the data support the conclusions?

Reviewer #1: Yes

Reviewer #2: Yes

3. Has the statistical analysis been performed appropriately and rigorously? 

Reviewer #1: Yes

Reviewer #2: Yes

4. Have the authors made all data underlying the findings in their manuscript fully available?

Reviewer #1: Yes

Reviewer #2: Yes

5. Is the manuscript presented in an intelligible fashion and written in standard English?

Reviewer #1: Yes

Reviewer #2: Yes

6. Review Comments to the Author

Reviewer #1: (No Response)

Reviewer #2: This manuscript, “An Insight into the Causal Relationship between Sarcopenia-Related Traits and Venous Thromboembolism: A Mendelian Randomization Study”, by Du et al introduced a Mendelian Randomization (MR) study to assess the causal effect of sarcopenia on the risk of VTE. While the results show that the SNPs genetically predicted sarcopenia did not causally influence VTE directly, this is a well-designed and interesting study. The paper was well written in general. I suggest that it be accepted.

7. PLOS authors have the option to publish the peer review history of their article (what does this mean?). If published, this will include your full peer review and any attached files.

Reviewer #1: **Yes: **Sarita Poonia

Reviewer #2: No

---

## [Editor Report · Acceptance letter]

29 Apr 2024

PONE-D-24-01767R1 

PLOS ONE

Dear Dr. Gu, 

I'm pleased to inform you that your manuscript has been deemed suitable for publication in PLOS ONE. Congratulations! Your manuscript is now being handed over to our production team.

Kind regards, 

on behalf of

Dr. Bo Hu 

Academic Editor

PLOS ONE